# Intermediate-phase-assisted low-temperature formation of γ-CsPbI₃ films for high-efficiency deep-red light-emitting devices

Chang Yi[1,4], Chao Liu[1,4], Kaichuan Wen[1], Xiao-Ke Liu [2], Hao Zhang[1], Yong Yu[2], Ning Fan[1], Fuxiang Ji[2], Chaoyang Kuang [2], Bo Ma[1], Cailing Tu[1], Ya Zhang[1], Chen Xue[3], Renzhi Li[1], Feng Gao [2✉], Wei Huang[1,3✉] & Jianpu Wang [1✉]

Black phase CsPbI₃ is attractive for optoelectronic devices, while usually it has a high formation energy and requires an annealing temperature of above 300 °C. The formation energy can be significantly reduced by adding HI in the precursor. However, the resulting films are not suitable for light-emitting applications due to the high trap densities and low photoluminescence quantum efficiencies, and the low temperature formation mechanism is not well understood yet. Here, we demonstrate a general approach for deposition of **γ**-CsPbI₃ films at 100 °C with high photoluminescence quantum efficiencies by adding organic ammonium cations, and the resulting light-emitting diode exhibits an external quantum efficiency of 10.4% with suppressed efficiency roll-off. We reveal that the low-temperature crystallization process is due to the formation of low-dimensional intermediate states, and followed by interionic exchange. This work provides perspectives to tune phase transition pathway at low temperature for CsPbI₃ device applications.

---

[1] Key Laboratory of Flexible Electronics (KLOFE) & Institute of Advanced Materials (IAM), Nanjing Tech University (NanjingTech), 30 South Puzhu Road, Nanjing 211816, China. [2] Department of Physics, Chemistry, and Biology (IFM), Linköping University, SE-58183 Linköping, Sweden. [3] Frontiers Science Center for Flexible Electronics (FSCFE), Shaanxi Institute of Flexible Electronics (SIFE), Xi'an Institute of Biomedical Materials & Engineering (IBME), Northwestern Polytechnical University (NPU), 127 West Youyi Road, Xi'an 710072, China. [4] These authors contributed equally: Chang Yi, Chao Liu. ✉email: feng.gao@liu.se; iamwhuang@nwpu.edu.cn; iamjpwang@njtech.edu.cn

All-inorganic halide perovskites such as CsPbX$_3$ (X = Cl, Br, I) are attractive materials for light emitters and photovoltaic applications due to their potential in overcoming long-term stability issues of organic–inorganic hybrid halide perovskites[1–5]. The low-temperature solution-processed phase-stable CsPbX$_3$ perovskites are mainly based on bromide and chloride with suitable tolerance factors[6]. Optically active CsPbI$_3$ black phases (α (cubic), β (tetragonal) or γ (orthorhombic)) usually require high annealing temperature (300–370 °C) to overcome the energy barrier for phase transition[7,8]. Moreover, the CsPbI$_3$ black phases can readily transform to yellow-phase δ-CsPbI$_3$ in ambient conditions due to the thermodynamic instability[9–11], inhibiting their application in optoelectronic devices[12,13]. In perovskite solar cells, a general method of forming black phases CsPbI$_3$ at low temperature is by adding hydroiodic acid (HI) in CsPbI$_3$ precursor solution prior to spin coating[13,14]. It has been observed that the judicious amount of HI would decompose the solvent dimethylformamide (DMF) to form dimethylammonium iodide[15], while the mechanism of how this process affects the crystallization of CsPbI$_3$ is still under intensive debate with two arguments. One argument is that the formed DMA would sublimate and lead to a fast crystallization of CsPbI$_3$[4,16,17]. Another argument is that the DMA becomes the part of the crystal structure and the formed black phase is not CsPbI$_3$ but Cs$_x$DMA$_{1-x}$PbI$_3$[18,19].

For CsPbI$_3$-based light-emitting diodes (LEDs) applications, the low-temperature HI doping method is difficult to achieve high performance devices, mainly due to the high trap density and strong nonradiative recombination with those perovskite films (typical photoluminescence quantum efficiency (PLQE) < 1%)[20]. Alternatively, high-efficiency LEDs has been demonstrated based on CsPbI$_3$ quantum dots (QDs)[21,22]. However, those colloidal QDs are synthesized ex situ in flasks by the hot-injection method, which usually requires a temperature above 170 °C and complicated processing conditions[21–24]. In addition, usually those perovskite QD-based LEDs only show high efficiency at low current densities with strong efficiency roll-off due to the strong nonradiative Auger process in perovskite QDs[21,25,26]. In this work, we report an effective approach for achieving high quality γ-CsPbI$_3$ at low annealing temperature (~100 °C) for high performance LEDs applications. More importantly, we reveal that the low-temperature formation process of black phase CsPbI$_3$ can be generally observed when intermediate states are formed, followed by an interionic exchange in the presence of large organic ammonium cations.

## Results

**Low temperature formed γ-CsPbI$_3$ films.** A DMF precursor solution of imidazolium iodide (IZI), CsI, and PbI$_2$ with a molar ratio of 4:1.5:1 (referred as IZI-CsPbI$_3$) is spin coated onto polyethylenimine ethoxylated (PEIE) modified ZnO substrates (referred as ZnO/PEIE). We note that ZnO/PEIE has been widely used as an electron transporting layer in perovskite LEDs[27]. After thermal annealing at 100 °C for 5 min, the IZI-CsPbI$_3$ film shows X-ray diffraction (XRD) peak of 14.3 and 28.9° without any splitting (Fig. 1a), corresponding to the (110) and (220) crystal planes of γ-CsPbI$_3$, respectively. An absorbance edge at ~1.75 eV and a photoluminescence (PL) peak at ~700 nm are also observed for this film (Fig. 1b). These results are consistent with the characteristics of γ-CsPbI$_3$ obtained through thermal annealing above the transition temperature (around 310 °C) and rapid cooling process[8,13]. In addition, the scanning electron microscopic (SEM) measurement shows that the IZI-CsPbI$_3$ film is discrete, consisting of particles with an average size of ~80 nm (Fig. 1c). The film shows good emission properties with PLQE reaching up to 38% (Fig. 1d). Time-correlated single photon counting measurement shows that the PL lifetime increases with the increasing amount of IZI (Supplementary Fig. 1a), suggesting that the nonradiative recombination of the γ-CsPbI$_3$ is suppressed with increasing IZI. This result is consistent with the PLQE result (Supplementary Fig. 1b). More importantly, the IZI-CsPbI$_3$ film exhibits negligible degradation after exposing for 36 days in ambient air at room temperature with 80% relative humidity (Fig. 1d). In contrast, the regular γ-CsPbI$_3$ obtained from the high-temperature annealing process can only retain for 4 h in the same environment (Supplementary Fig. 2). The PL intensity of IZI-CsPbI$_3$ film dropped to 50% over 8 days in the ambient air (Supplementary Fig. 3), suggesting significantly improved phase and optical stability compared to previously reported results[4,19].

**The mechanism behind low temperature formed γ-CsPbI$_3$ films.** To investigate the mechanism of IZI on facilitating the formation of γ-CsPbI$_3$ perovskite at the low temperature, we monitor the crystal phase evolution of the as-spun IZI-CsPbI$_3$ film by XRD measurements under various annealing time (Fig. 2a). At early stage of the thermal annealing process (10–15 s), CsI with the peak at 27.6° remains unchanged, while an intermediate phase with peaks at 11.3 and 25.4° is formed. This intermediate phase can be assigned to one dimensional (1D) IZPbI$_3$, since their XRD peaks are consistent (Fig. 2a red line and Supplementary Table 1). Upon further annealing (~30–60 s), both the XRD peaks (11.3, 25.4, and 27.6°) of intermediate phase IZPbI$_3$ and CsI disappear. In the meantime, XRD peaks (14.3 and 28.9°) of γ-CsPbI$_3$ perovskite appear. These facts suggest that the 1D IZPbI$_3$ perovskite transforms to the γ-CsPbI$_3$ perovskite during the low-temperature annealing process. This transformation process requires an interionic exchange process of IZ$^+$ embedded in face-shared PbI$_6$ chains with external Cs$^+$. The corresponding morphology of IZI-CsPbI$_3$ films annealed at 100 °C for various time durations is also monitored by SEM measurement (Supplementary Fig. 4). The unannealed film displays a dense, planar morphology (Supplementary Fig. 4a). With a short-time annealing ($t = 10$ s), mounts of small grains of about 40 nm emerge (Supplementary Fig. 4b), corresponding to the intermediate phase. By extending the annealing time duration to 15 s, the small grains grow bigger and the layer becomes discrete (Supplementary Fig. 4c), corresponding to the mixed phase with 1D and 3D. When annealed over 30 s, the discrete γ-CsPbI$_3$ grains with an average size of ~80 nm form and disperse on the ZnO/PEIE substrate (Supplementary Fig. 4d–f).

We find that without the underneath ultrathin PEIE layer, the low-temperature phase transformation can be still observed by XRD measurement (Supplementary Fig. 5). It cannot be formed at 100 °C without ZnO layer, where both the 1D phase and CsI remain unchanged even after 10 min annealing (Supplementary Fig. 6). UV–vis absorption spectra measurement result is consistent with the above XRD result (Supplementary Fig. 7). These results suggest that during the formation of γ-CsPbI$_3$ from the intermediate phase IZPbI$_3$, the ZnO substrate plays important roles in the interionic exchange process.

We then use X-ray photoelectron spectroscopy (XPS) measurement to reveal the role of the ZnO substrate in the interionic exchange process. Particularly, the chemical interaction between the films (IZI, IZI-CsPbI$_3$) and ZnO is investigated. Figure 2b shows the high resolution XPS spectra of N 1$s$, O 1$s$, and Zn 2$p$ of these films. The XPS spectrum of IZI prepared on ITO substrate shows two N 1$s$ peaks at around 401.7 eV (N-3) and 400.2 eV (N-1), respectively (Fig. 2b black and pink line). When IZI film is on top of ZnO layer, the N-3 peak disappears and a new peak

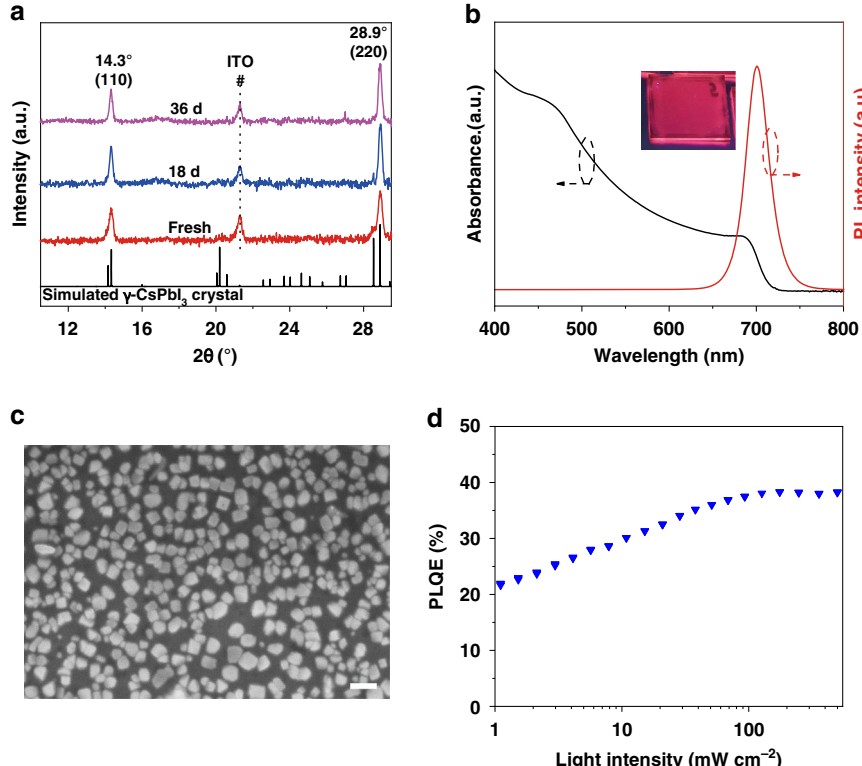

**Fig. 1 Characteristics of IZI-CsPbI₃ film on ZnO/PEIE substrate. a** XRD pattern after exposed in the air with 80% relative humidity for various durations. Black line is the simulated powder patterns of γ-CsPbI₃[8]. **b** UV–vis and PL spectrum. Inset shows the image under UV light. **c** SEM image (scale bar, 200 nm), and **d** Excitation-intensity-dependent PLQE.

positioning at a lower binding energy of 398.8 eV shows up. The two peaks of the ZnO/IZI film are in well agreement with those of imidazole (IZ), and the peak at 398.8 eV can be attributed to the N-2 of IZ (Fig. 2b blue line)[28,29]. The above XPS result indicates that the ZnO can deprotonate the IZ⁺ cation of IZI. Importantly, this deprotonation process can be also observed in the IZI-CsPbI₃ film with ZnO and ZnO/PEIE substrate, respectively (Fig. 2b and Supplementary Fig. 8). Moreover, compared to the bare ZnO film on ITO, the ZnO/IZI and ZnO/IZI-CsPbI₃ films show significantly suppressed XPS peak at 530 eV ascribed to O 1s of Zn–O bonding (Fig. 2c black line)[30,31], and their peaks of the Zn 2p of Zn–O bonding shift to higher energies (Fig. 2d). These further XPS observations can be explained by the presence of H⁺ in the Zn–O–H bonding which decreases electron cloud density around Zn atoms[32], consistent with the result of deprotonation of the IZ⁺ cation by ZnO. Figure 2e shows a scheme of this deprotonation process of IZ⁺ by ZnO. In addition, attenuated total reflection Fourier-transform infrared (ATR-FTIR) spectroscopy measurement result can further confirm this scenario. As shown in Supplementary Fig. 9, the N–H stretching vibration at 3260 cm⁻¹ of IZI in the films on the ZnO/PEIE substrate is shifted to higher wavenumber (3330 cm⁻¹). Correspondingly, the broad stretching vibration of Zn–O is red shifted from 556 to 538 cm⁻¹[33]. And the signature peak of IZ at about 3130 (CH stretching), 1543 (NH bend), 1328 (CH bend), 1263 (ring breathing), 1055 (CH bend), 841 (ring bend), 757 (CH out-of-plane bend), and 658 (torsion) cm⁻¹ are clearly observed[34]. We note that the similar deprotonation process can also be observed in films with the intermediate phase of IZPbI₃ (Supplementary Fig. 10) and FAPbI₃ perovskites[35].

On the basis of the above phase evolution and chemical elementary analysis, we can have a clear picture of the mechanism

of the γ-CsPbI₃ formation process at low temperature, as shown in Fig. 2f. It first forms an intermediate phase IZPbI₃, followed by the formation of γ-CsPbI₃ through the interionic exchange of IZ⁺ with external Cs⁺ in the process of the deprotonation of IZ⁺ with ZnO. The overall chemical reaction of phase formation can be presented as follows:

Stage 1 : $PbI_2 + C_3N_2H_5I + Cs^+ \rightarrow (C_3N_2H_5)PbI_3 + Cs^+$ forming intermediate phase,

Stage 2 : $2(C_3N_2H_5)PbI_3 + 2Cs^+ + 2ZnO \rightarrow 2CsPbI_3 + 2C_3N_2H_4 + Zn(OH)_2 + Zn^{2+}$ ion exchange and forming $\gamma-CsPbI_3$.

Since the ZnO induced deprotonation process likely mainly occurs at the ZnO interface, it is interesting to investigate how thick the perovskite can be formed by this approach. Supplementary Fig. 11 shows the absorbance of the films with various thickness fabricated from different concentration of precursor solutions. It shows the absorbance at 687 nm from the black phase CsPbI₃ increases linearly with the film thickness <200 nm. Above 200 nm, the absorbance saturates and declines. This result suggests that the ZnO substrate can facilitate the γ-CsPbI₃ formation within the film thickness of 200 nm.

**Kinetic of phase transition.** We further investigate the crystallization kinetics of the γ-CsPbI₃ films with various IZI contents by using time-dependent UV–vis spectroscopy. The reaction progress of these films with different contents of IZI in precursor solutions (IZI/PbI₂ molar ratio is x, x = 0, 0.5, 1, 2, 3, 4, and 5) were monitored through change in absorbance ($A(t)$) at ~687 nm (Fig. 1b). The value of formed γ-phase fraction $\chi(t)$ is defined as:

$$\chi(t) = \frac{A(t)}{A(t_{end})}, \quad (1)$$

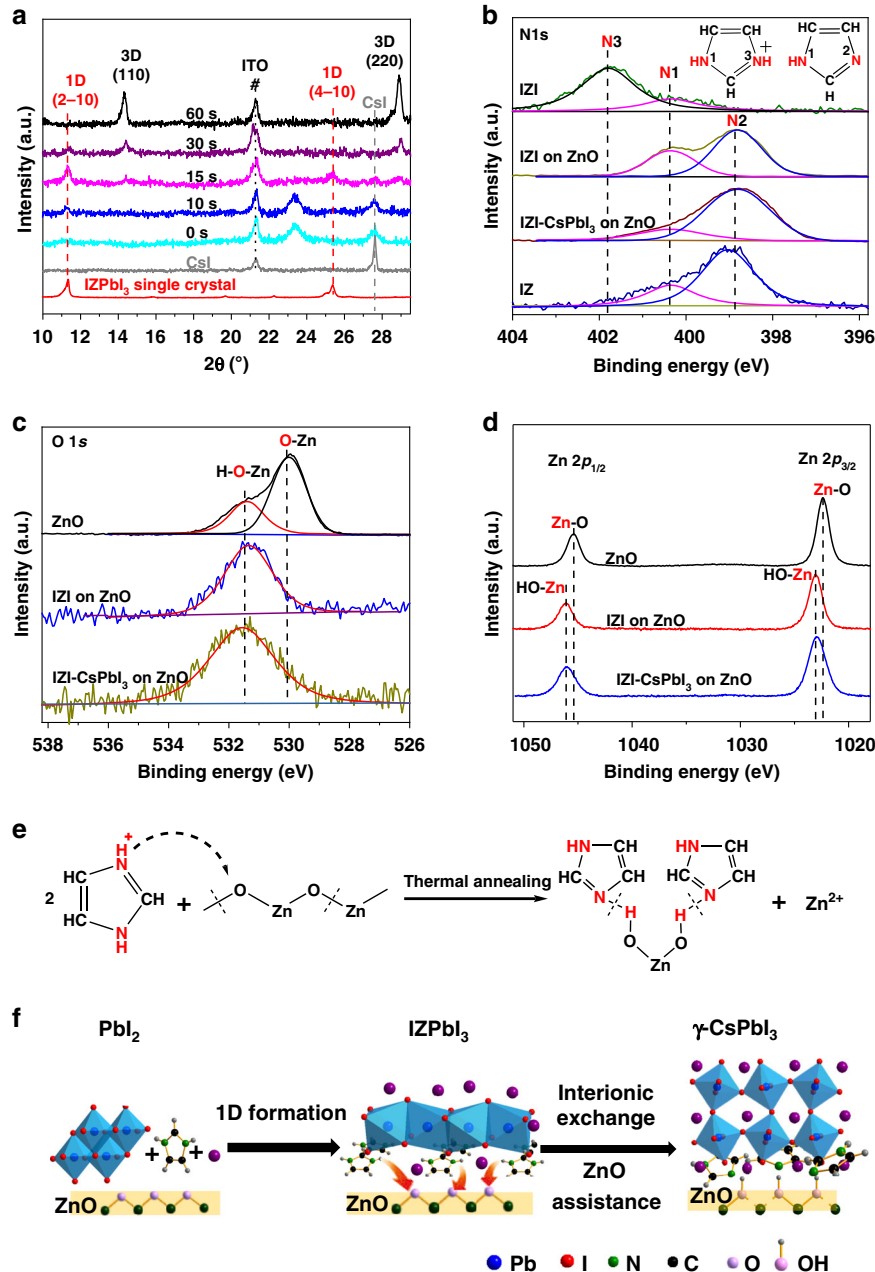

**Fig. 2 Formation process of γ-CsPbI₃ perovskite on ZnO/PEIE substrate. a** XRD patterns of IZI-CsPbI₃ film on ZnO/PEIE substrate with different annealing time at 100 °C. XPS spectra of (**b**) N 1s, (**c**) O 1s, (**d**) Zn 2$p_{1/2}$ and Zn 2$p_{3/2}$ for IZI, IZI-CsPbI₃ film on ZnO substrate and for IZI, IZ film on ITO substrate, respectively. **e** Schematic diagram of deprotonation of IZ⁺ with ZnO. **f** Mechanism of the γ-CsPbI₃ formation process at low temperature on ZnO substrate.

where the $A(t_{end})$ represents the maximum absorbance at the final state. As shown in Fig. 3a, the rate of γ-phase formation increases with the increasing amount of IZI at a constant annealing temperature of 90 °C. And Fig. 3b shows that the rate also increases with increasing annealing temperature when the IZI content is constant. More detailed measurement results on different mole ratios and various annealing temperature are shown in Supplementary Fig. 12. The activation energy of the γ-CsPbI₃ formation for various IZI contents can be estimated by using the Mitte-meijer model[36], as shown in Supplementary Fig. 12g. Figure 3c shows a summary of the estimated activation energies barrier, which can be significantly decreased from 150 to 29 kJ mol⁻¹ with increasing molar ratio of IZI to PbI₂ from 0.5 to 5.

**Generality of the low temperature formed γ-CsPbI₃ films.** In order to further demonstrate the generality of the low-temperature formation of black CsPbI₃ through the intermediates, we add various RNH₃⁺-based large organic cations, such as butylammonium iodine (BAI), hexylammonium iodine (HAI), phenethylammonium iodine (PEAI), and naphthylethylammonium iodine (NMAI) into the CsPbI₃ precursor and spin coated on top of ZnO substrates. As shown in Supplementary Fig. 13, all those films exhibit formation of intermediates, decomposition, and interionic exchange during the annealing process. And finally, the black CsPbI₃ films form at 100 °C. Therefore, we believe that the process we plotted in Fig. 2f is general for forming black CsPbI₃ at low temperature assisted by RNH₃⁺-based large organic cations.

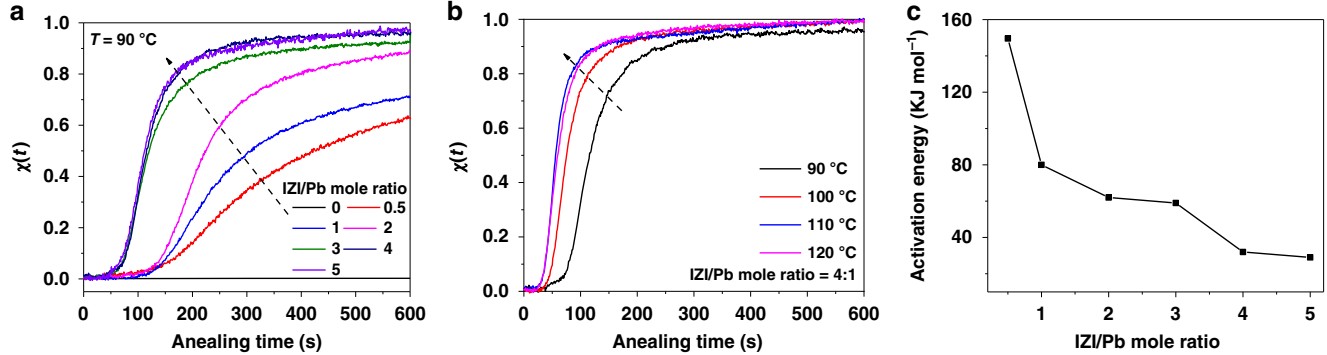

**Fig. 3 Kinetic data for isothermally annealed IZI-CsPbI$_3$ films on ZnO/PEIE substrate.** Influence of annealing time on the γ-phase transition fraction $\chi(t)$ for (**a**) the IZI-CsPbI$_3$ films with different mole ratio (0:1, 0.5:1, 1:1, 2:1, 3:1, 4:1, 5:1) at 90 °C, and for (**b**) IZI-CsPbI$_3$ with mole ratio of 4:1 at different temperature. $\chi(t)$ is defined as $A(t)/A(t_{end})$, $A(t)$ represents the time-dependent absorbance at ~687 nm, $A(t_{end})$ represents the max absorbance. **c** The dependence of activation energies of the γ-CsPbI$_3$ films on the IZI content.

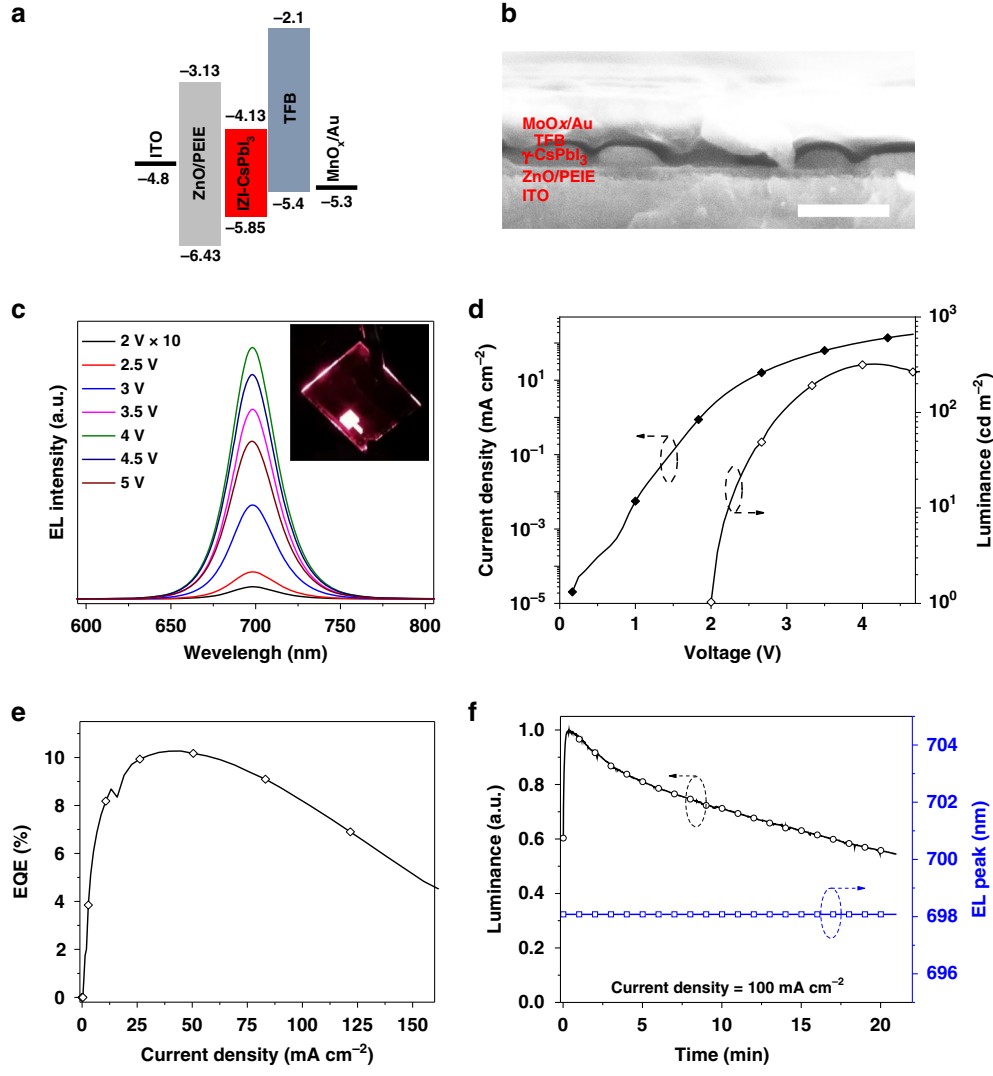

**Fig. 4 IZI-CsPbI$_3$ LED structure and optoelectronic characteristics. a** Flat-band energy level diagram and **b** Cross-sectional SEM image (scale bar, 200 nm). **c** EL spectra under various voltages. Inset shows the photograph of the deep-red device. **d** Current density and luminance versus voltage (*J–L–V*). **e** EQE versus current density (*EQE–J*) curves. **f** Lifetime and the maximum EL peak position measurement at 100 mA cm$^{-2}$. The maximum EQE as the initial value to calculate the $T_{50}$.

**LED device performance**. As shown in Fig. 4a, the LED devices have a structure of ITO/ZnO/PEIE/IZI-CsPbI$_3$/poly (9,9-dioctyl-fluorene-co-N-(4-butylphenyl)diphenylamine) (TFB)/molybdenum oxide (MoO$_x$)/gold (Au). Detailed fabrication process can be found in "Methods." The valence bands of IZI-CsPbI$_3$ film is obtained via ultraviolet photoelectron spectroscopy measurement (Supplementary Fig. 14), and the conduction band was estimated using the band gap derived from the absorption band edge (Fig. 1b). A cross-sectional image of the device shows the formation of the discrete CsPbI$_3$ particles layer with a thickness of ~40 nm (Fig. 4b). We note that the residual IZI can locate between the discrete particles, which can prevent the device from short-circuit current and enhance the light-outcoupling efficiency, and similar studies have been demonstrated early[37–39]. The current density and luminance versus voltage ($J$–$L$–$V$) and EQE curves of the device with different contents of IZI are shown in Supplementary Fig. 15. The electroluminescence (EL) peak is located at ~698 nm and the shape remains unchanged under different bias voltages (Fig. 4c). The champion device based on the IZI-CsPbI$_3$ film (mole ratio of 4:1.5:1 for IZI:CsI:PbI$_2$) exhibits a peak EQE of 10.4% with luminance of 340 cd m$^{-2}$, and the turn-on voltage is as low as 2 V (Fig. 4d, e). We note that our device peak EQE is a record for 3D CsPbI$_3$ film-based red LEDs[40,41]. Moreover, the efficiency roll-off of the device is significantly suppressed, and the EQE remains high (~8%), under a high current density of 100 mA cm$^{-2}$ (Fig. 4e). This feature is very different compared with previous QD-based CsPbI$_3$ LED devices[21,26], where high EQE can only be obtained at low excitations likely due to the strong nonradiative Auger process. In addition, the IZI-CsPbI$_3$ devices exhibit highly reproducible with average EQE of 8.1% for 75 devices (Supplementary Fig. 15c). The best device shows a half-lifetime of 20 min at a constant current density of 100 mA cm$^{-2}$, and the EL peak position remains constant over time (Fig. 4f).

## Discussion

In summary, we have developed a low-temperature method of forming CsPbI$_3$ black phases for high performance CsPbI$_3$ LED applications via synergistic effect of IZI and ZnO electron transport layer. The judicious amount of IZ$^+$ in the precursor promotes the intermediate phase formation, followed by the formation of γ-CsPbI$_3$ through the interionic exchange of IZ$^+$ with external Cs$^+$ in the process of the deprotonation of IZ$^+$ with ZnO. The phase transition engineering can efficiently reduce the formation energy of CsPbI$_3$ black phase, and facilitate the formation of discrete CsPbI$_3$ particles film with high PLQE and long-term stability. The resulting CsPbI$_3$ LED shows a peak EQE of 10.4% with suppressed efficiency roll-off. Importantly, the low-temperature formation process can be generally observed with various RNH$_3$$^+$-based large organic cations. So we believe that our work provides useful perspectives to tune the phase transition pathway, and offers an effective approach to fabricate low-temperature processed CsPbI$_3$ black phase film for LEDs applications.

## Methods

**Synthesis of ZnO colloidal solution**. ZnO were synthesized by following the previously reported method[27]. The dimethyl sulfoxide solution of Zn(Ac)$_2$·2H$_2$O (3 mmol in 30 mL) was mixed in ethanol solution of tetramethylammonium hydroxide pentahydrate (TMAH·5H$_2$O) (5.6 mmol in 10 mL) and stirred at 30 °C for 24 h. The ZnO colloids were precipitated with ethyl acetate, and washed it three times with ethanol and ethyl acetate. Finally, the obtained ZnO colloid were dispersed in ethanol and set aside in the fridge until serve.

**Synthesis of organic ammonium salt**. IZI was prepared by mixing IZ (2 g) and excess hydroiodic acid (45 wt% in water) in 15 mL of ethanol at 0 °C. After the reaction mixture was stirred for 2 h, 60 mL diethyl ether was added into the mixture to obtain the precipitates. The collected precipitates were washed three

times with diethyl ether and stored in an oven. BAI, HAI, PEAI, and NMAI were prepared by similar method.

**Perovskite precursor solutions preparation**. The CsPbI$_3$ precursor solution were prepared by dissolving IZI, CsI, and PbI$_2$ with a molar ratio of $x$ in DMF at weight percent (wt%) of 6% (IZI and PbI$_2$ molar ratio is $x$:1, $x$ = 0, 1, 2, 3, 4, CsI and PbI$_2$ molar ratio fixed at 1.5:1), and the solution with molar ratio of $x$ = 4 is referred as IZI-CsPbI$_3$. The IZI–PbI$_2$ precursor solutions were prepared by dissolving IZI and PbI$_2$ with a molar ratio of 4:1 in DMF. BAI-CsPbI$_3$, HAI-CsPbI$_3$, PEAI-CsPbI$_3$, and NMAI-CsPbI$_3$ precursor solution were prepared by dissolving BAI, HAI, PEAI, NMAI in DMF solution of CsI and PbI$_2$ with molar ratio of 4:1.5:1, 2:1.5:1, 4:1.5:1, 2:1.5:1, respectively.

**Perovskite film deposition**. ZnO colloidal solution was deposited onto the ITO substrate using spin-coating technique at 4000 rpm for 45 s, followed by annealing at 150 °C for 30 min The PEIE (1.5 mg mL$^{-1}$ in methoxyethanol) was spin coated onto the ZnO films at a speed of 5000 rpm and annealed at 100 °C for 10 min. Finally, the precursor solution was spin coated onto ITO/ZnO/PEIE substrate (4000 rpm, 30 s) or ITO substrate with 100 °C annealing for various time to form films, respectively.

**Device fabrication**. The devices were fabricated with a structure of ITO/ZnO/PEIE/perovskite/TFB/MoOx/Au. After the deposition of the ZnO and perovskite films as mentioned above, the TFB (12 mg mL$^{-1}$ in chlorobenzene) layer was spin coated by 3000 rpm for 30 s. Finally, MoO$_x$ (7 nm) and Au (80 nm) were deposited by thermal evaporation, respectively

**Perovskite film characterization**. XRD measurements were performed with a Rigaku Smart lab (3 kW) XRD patterns with Bragg–Brentano focusing, a diffracted beam monochromator and a conventional Cu target X-ray tube set to 40 kV and 30 mA. Time-dependent UV–vis absorption spectra were obtained on PerkinElmer Lambda 950 spectrometer. The general morphologies of the films were characterized by FEI (Quanta 200 FEG) SEM under a voltage of 5 kV. XPS tests were carried out using a Thermo ESCALAB250 Xi X-ray photoelectron spectrometer with Al K$_\alpha$ X-ray as the excitation source. All binding energies were referred to the C 1s peak at 284.8 eV of the surface areas of the samples. ATR-FTIR spectra of films were characterized by a Thermo-Niclet IS50 equipped with a Smart SAGA reflectance accessory in the range of 450–4000 cm$^{-1}$. PL spectra were obtained using a fluorescent spectrophotometer (F-4600, HITACHI) with a 200 W Xe lamp as an excitation source. The Excitation-intensity-dependent PLQE of perovskite films was monitored by a joint control of a 450 nm continuous wave laser, 1000 μm slit width, optical fiber spectrometer, and integrating sphere[42]. The film thickness was determined by a surface profiler (KLA-Tencor).

**Kinetic modeling of the phase transition**. The dependence of rate on temperature indicates a significant activation energy barrier ($E_a$) for the process of the γ-CsPbI$_3$ formation. We worked out the activation energy barrier using the Mittemeijer model:[36]

$$\ln(t_{x2} - t_{x1}) = \frac{E_a}{RT} - \ln k_0 + \ln(\beta_{x2} - \beta_{x1}), \qquad (2)$$

where $E_a$ is the effective activation energy barrier, $t_{x1}$ and $t_{x2}$ are the annealing time at which the transformed fraction is $\chi(t) = 0.2$ and 0.8, $R$ is the gas constant, $T$ is the temperature, and $k_0$ is a rate constant prefactor.

**Device characterization**. The LED was measured in glove box at room temperature, and detailed setup can be found in reference[38]. A Keithley 2400 source meter with a step of 0.05 V s$^{-1}$ and a fiber integration sphere (FOIS-1) coupled with a QE65 Prospectrometer were used for the device measurements. The device area is 7.25 mm$^2$. The device lifetime was measured by using the same setup under a constant current density of 100 mA cm$^{-2}$.

## Data availability

The data that support the finding of this study are available from the corresponding author upon reasonable request.

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

## Acknowledgements

This work is financially supported by the Major Research Plan of the National Natural Science Foundation of China (91733302), the National Natural Science Foundation of China (51703094, 61935017, 61974066), the Natural Science Foundation of Jiangsu Province, China (BK20170991), the National Science Fund for Distinguished Young Scholars (61725502), Major Program of Natural Science Research of Jiangsu Higher Education Institutions of China (18KJA510002), the National Key Research and Development Program of China (2018YFB0406704), and the Natural Science Fund for Colleges and Universities in Jiangsu Province of China (17KJB150023). The work at Linköping is funded by the ERC Starting Grant (717026) and the Swedish Government Strategic Research Area in Materials Science on Functional Materials at Linköping University (Faculty Grant SFO-Mat-LiU no. 2009-00971). X.-K.L. is a Marie Skłodowska-Curie Fellow (No. 798861). F.G. is a Wallenberg Academy Fellow. We thank Dr. Xianjie Liu for XPS measurement and analysis. We thank Dr. Zhangjun Hu for FTIR measurement and analysis. Open Access funding provided by Linköping University.

## Author contributions

J.W. and C.Y. had the idea for and designed the experiments. J.W., W.H., and F.G. supervised the work. C.Y. and C.L. carried out the device fabrication and characterizations and films characterizations. K.W. and B.M. carried out the PLQE characterizations. The ZnO was synthesized with the assistance of C.K. and N.F. The single crystal of IZPbI₃ was cultivated with the assistance of F.J. and C.X. C.T., Y.Z., and H.Z. conducted the SEM measurements. Y.Y. conducted the XPS measurement. C.Y. wrote the paper. J.W., F.G., W.H., X.-K.L., and R.L. provided revisions. All authors discussed the results and commented on the paper.

## Competing interests

The authors declare no competing interests.
