## [Peer Review File · Nature Communications]

Reviewers' comments:

Reviewer #1 (Remarks to the Author):

J. Wang et al report the fabrication of black phase CsPbI₃ at low annealing temperature of 100 °C with high photoluminescent quantum efficiency (PLQY) for light-emitting diodes (LEDs) application. The use of organic ammonium imidazolium iodide (IZI) precursor and zinc oxide (ZnO) electron transport layer enables to control the phase transition. The IZI-CsPbI₃ film shows PLQY of 38% and negligible crystal degradation even after exposing 36 days in the air. The LED based on IZI-CsPbI₃ film shows a peak external quantum efficiency (EQE) of 10.4% with good reproducibility. The addition of IZI into perovskite played a role for stable crystal structure and LED efficiency. However, the discussions and results regarding the optoelectrical properties such as optical stability, PL decay time, and energy levels (valence/conduction band) are insufficient in this manuscript. In addition, a lot of papers regarding the high efficiency red perovskite LEDs with EQE of over 10% have already been reported. Therefore, this manuscript is not suitable for publication in this stage.

1. The inverted perovskite LEDs with ZnO and amine polymers have been demonstrated to improve PLQY and EQE (for example; J. Phys. Chem. Lett. 2016, 7, 4602 and Adv. Mater. 2018, 30, 1804691). In this manuscript, the PLQY of IZI-CsPbI₃ film was 38%, which not higher compared to the conventional CsPbI₃ films.

2. In order to confirm the effect of ZnO in the interionic exchange process, the authors should be added the reference sample without ZnO layer.

3. The explanation of PEIE layer is insufficient because ZnO layer plays role in intermediate phase transition. The use of PEIE layer disturbs a direct contact between ZnO and IZI-CsPbI₃ film.

4. In this manuscript, various experimental data were provided regarding the crystal structure characterization of IZI-CsPbI₃ film. However, there is no information about the PL decay profile, which is very important data to discuss an optical property of IZI-CsPbI₃ film.

5. The energy levels such as valence band and conduction band of perovskite samples were not described in this manuscript.

6. The authors showed a crystal stability test by XRD measurement. The optical stability test of IZI-CsPbI₃ film is also important.

Reviewer #2 (Remarks to the Author):

The authors report efficient CsPbI₃ light-emitting diodes by employing an imidazolium additive. The authors have also performed thorough experimental studies to determine the mechanism of low temperature CsPbI₃ perovskite formation through the use of imidazolium. Given that cesium-based perovskites show good promise in long-term device stability, this work is important and could be published in nature communications upon minor revision.

1) Could the authors comment if it is experimentally confirmed that the imidazole remains in the perovskite film/layer? Presumably this could be determined by some spectroscopic techniques.

2) The authors mentioned that other large ammonium salts worked as well. How crucial is the identity of this ammonium salt for general device performance?

3) The imidazole has helped in the formation of CsPbI₃ at low temperature. Could the authors comment on whether it is helpful for keeping CsPbI₃ in its black phase over the long term? How does it compare to CsPbI₃ formed at high temperature without the help of imidazole?

Reviewer #3 (Remarks to the Author):

The authors have developed a low-temperature fabrication method of γ -CsPbI₃, where organic ammonium cations were introduced (through imidazolium iodide (IZI)) to form an intermediate phase to mediate the formation process of the perovskite. Through XPS and ATR-FTIR characterizations, the authors pointed out that the deprotonation of the IZ⁺ cations by ZnO promotes interionic exchange between IZ⁺ and Cs⁺, and therefore facilitates the formation of the γ -CsPbI₃. The work also demonstrated a record EQE of 10.4% in 3D CsPbI₃-based red PeLEDs. The findings reported in this manuscript are interesting and may have important impact on low-temperature fabrication of thin film all-inorganic perovskites. In general, this is a high-quality work. I would recommend acceptance of this manuscript after the following issues are properly addressed.

1. Figure 2b shows different N 1s peaks of IZI on ZnO and on ITO. It is suggested that ZnO can deprotonate the IZ⁺ cation of IZI through formation of Zn-O-H bonds. In principle, ITO should also allow for a similar deprotonation process through formation of Sn-O-H bonding. Why does the latter process not occur? Can the authors please clarify what properties are required for the metal oxide to initiate the deprotonation process?
2. Regarding the mechanism illustrated in Figure 2f: if the interionic exchange is enabled by ZnO, which contacts the bottom surface of the perovskite, could the exchange happen efficiently at the top surface of the perovskite? Does the thickness of the perovskite film impose a limit to the effectiveness of this method?
3. It might be helpful if the authors could provide some morphological characterizations to assist understanding of the perovskite formation process.
4. For the stage 2 of chemical reaction, Zn²⁺ ions are formed. Will these ions induce trap states?
5. Ammonium salts are reported to passivate defects in perovskite films and at interfaces in PeLEDs, leading to promotion of device performance and lifetime. In this regard, could the addition of IZI also lead to improvement in film quality or interfaces and therefore better LED performances? Also, how about stability?

Point-by-Point Response to Referees

Reviewer #1 (Remarks to the Author):

Comment #1: J. Wang et al report the fabrication of black phase CsPbI₃ at low annealing temperature of 100°C with high photoluminescent quantum efficiency (PLQY) for light-emitting diodes (LEDs) application. The use of organic ammonium imidazolium iodide (IZI) precursor and zinc oxide (ZnO) electron transport layer enables to control the phase transition. The IZI-CsPbI₃ film shows PLQY of 38% and negligible crystal degradation even after exposing 36 days in the air. The LED based on IZI-CsPbI₃ film shows a peak external quantum efficiency (EQE) of 10.4% with good reproducibility. The addition of IZI into perovskite played a role for stable crystal structure and LED efficiency. However, the discussions and results regarding the optoelectrical properties such as optical stability, PL decay time, and energy levels (valence/conduction band) are insufficient in this manuscript. In addition, a lot of papers regarding the high efficiency red perovskite LEDs with EQE of over 10% have already been reported. Therefore, this manuscript is not suitable for publication in this stage.

Response: We thank the reviewer for the constructive comments and useful suggestions, which have helped us to improve the manuscript.

Although red perovskite LEDs with EQEs of over 10% have already been reported, these devices were all used the colloidal CsPb(Cl/Br/I)₃ quantum dots (QDs) as emitters (*Nat. Photonics* **12**, 681-687 (2018). *Adv. Mater.* **30**, 1804691 (2018). *ACS Energy Lett.* **3**, 1571–1577 (2018). *Nano Lett.* **20**, 2829-2836 (2020)). Those colloidal QDs are synthesized ex-situ in flasks by the hot-injection method, which usually requires a temperature above 170 °C and complicated processing conditions. In addition, usually those perovskite QD based LEDs only show high efficiency at low current densities with strong efficiency roll-off due to the strong non-radiative Auger process in perovskite QDs (*Nat. Photonics* **12**, 681-687 (2018). *Adv. Mater.* **30**, 1804691 (2018). *ACS Energy Lett.* 2018, **3**, 1571–1577. *J. Am. Chem. Soc.* **140**, 562-565 (2017)). Therefore, in situ preparing CsPbI₃ film on charge transporting substrate at low temperature for high performance red LEDs is still a major challenge. We have clarified this in the revised manuscript

(Line 57 to 62, Page 2, highlighted).

In this work, we have successfully fabricated high-efficiency red perovskite LEDs based on in situ prepared CsPbI₃ films at low temperature. More importantly, we have clearly demonstrated the mechanism behind and shown how to form the optically-active γ -CsPbI₃ through tuning the phase transition pathway and overcoming the energy barrier at low temperature ($\sim 100^\circ\text{C}$). These in situ solution-processed γ -CsPbI₃ film-based LEDs exhibits an EQE of 10.4 % with suppressed efficiency roll-off and color stability under a high current density of 100 mA cm^{-2} . We believe that our work provides new perspectives to tune phase transition pathway at low temperature for CsPbI₃ based applications.

Regarding the extra measurement required, we have included additional data in the revised manuscript, such as optical stability, PL decay time, and energy levels, which can be found in detail below.

Comment #2: The inverted perovskite LEDs with ZnO and amine polymers have been demonstrated to improve PLQY and EQE (for example; *J. Phys. Chem. Lett.* 2016, 7, 4602 and *Adv. Mater.* 2018, 30, 1804691). In this manuscript, the PLQY of IZI-CsPbI₃ film was 38%, which not higher compared to the conventional CsPbI₃ films.

Response: We agree that the ZnO/PEI electron transport layer has been introduced to the perovskite LED community from the beginning to reduce the defect states and enhance EQE. As we mentioned in the Response to the Comment #1, the conventional CsPbI₃ films are fabricated from colloidal nanocrystals, which need extra synthesis with much higher temperature and suffering from strong Auger recombinations. The low-temperature in situ solution-processed CsPbI₃ film are still rarely reported at present due to their high formation energy and thermodynamic instability. Therefore, the 38% PLQE from low-temperature in-situ formation perovskite with suppressed Auger effect represent an important advance in red perovskite emitters.

Comment #3: In order to confirm the effect of ZnO in the interionic exchange process, the authors should be added the reference sample without ZnO layer.

Response: The crystal phase evolution of the reference sample without ZnO layer as shown in **Supplementary Fig. 6**. The low temperature phase transformation cannot be observed without ZnO

layer, where both the 1D phase and CsI remain unchanged even after 10 min annealing. UV-vis absorption spectra measurement result is consistent with the above XRD result (**Supplementary Fig. 7**). This result suggests that during the formation of γ -CsPbI₃ from the intermediate phase IZPbI₃, the ZnO substrate plays important roles in the interionic exchange process. (Line 120 to 125, Page 4-5, highlighted, **Supplementary Fig. 7 black line**).

Supplementary Fig. 6 (a) XRD patterns of IZI-CsPbI₃ film on ITO substrate with different annealing time at 100 °C. (b) Schematic diagram of the phase evolution of IZI-CsPbI₃ film on ITO substrate. Note that γ -CsPbI₃ cannot be formed without ZnO layer, and both the 1D phase (intermediate) and CsI remain unchanged.

Supplementary Fig. 7 UV-vis spectrum of IZI-CsPbI₃ film on ITO/PEIE substrate (black line), ZnO substrate (red line) and ZnO /PEIE substrate (blue line), respectively. Note that γ -CsPbI₃ cannot be formed without ZnO layer.

Comment #4: The explanation of PEIE layer is insufficient because ZnO layer plays role in intermediate phase transition. The use of PEIE layer disturbs a direct contact between ZnO and IZI-CsPbI₃ film.

Response: We compared the crystal phase evolution of IZI-CsPbI₃ film on ZnO layer with and without the underneath ultrathin PEIE layer, we find that there is no difference between on ZnO/PEIE and on ZnO for phase transformation (**Fig. 2b, c and Supplementary Fig. 5**). The absorption shoulder of γ -CsPbI₃ phase at around 687 nm are observed for IZI-CsPbI₃ film prepared on ZnO/PEIE substrate (Line 121 to 123, Page 5, highlighted, **Supplementary Fig. 7 blue line**). The XPS result indicates that this deprotonation of IZ⁺ can be also observed in the IZI-CsPbI₃ film with ZnO and ZnO/PEIE substrate, respectively (Line 136 to 138, Page 5, highlighted, **Fig.2b and Supplementary Fig. 8**). The ATR-FTIR spectroscopy measurement results are consistent with the XPS results (**Supplementary Fig. 9**). These results suggest that the ultrathin PEIE layer dose not completely prevent the direct contact and interactions between ZnO and perovskite precursor film.

Fig. 2a XRD patterns of IZI- CsPbI₃ film on ZnO /PEIE substrate with different annealing time at 100 °C

Supplementary Fig. 5 XRD patterns of IZI-CspbI₃ film on ZnO substrate without the underneath ultrathin PEIE layer under various annealing time at 100 °C.

Supplementary Fig. 7 UV-vis spectrum of IZI-CsPbI₃ film on ITO/PEIE substrate (black line), ZnO substrate (red line) and ZnO /PEIE substrate (blue line), respectively. Note that the ultrathin PEIE layer dose not prevent the direct contact and interactions between ZnO and perovskite precursor film.

Fig. 2 XPS spectra of N 1s (b), O 1s (c) for IZI, IZI-CsPbI₃ film on ZnO substrate and for IZI, IZ film on ITO substrate, respectively.

Supplementary Fig. 8 XPS spectra of (a) N 1s and (b) O 1s for IZI-CsPbI₃ film on ZnO and ZnO/PEIE substrate, 6 / 19

respectively. Note that the underneath ultrathin PEIE does not prevent the deprotonation of the IZ⁺ cation by ZnO.

Supplementary Fig. 9a ATR-FTIR spectra of IZI-CsPbI₃ films on ZnO and ZnO/PEIE substrate, respectively. The signature peak of imidazole (IZ) at about 3345 ($\nu_{\text{N-H}}$), 3130 ($\nu_{\text{C-H}}$), 1543 (δ_{NH}), 1328 (δ_{CH}), 1263 (ring breathing), 1055 (δ_{CH}), 841 (ring bend), 757 (γ_{CH}) and 658 (torsion) cm^{-1} are clearly observed on ZnO/PEIE substrate, indicating the IZ⁺ cation could also be deprotonated by ZnO/PEIE to form IZ.

Comment #5: In this manuscript, various experimental data were provided regarding the crystal structure characterization of IZI-CsPbI₃ film. However, there is no information about the PL decay profile, which is very important data to discuss an optical property of IZI-CsPbI₃ film.

Response: We thank the reviewer for the constructive comments. We have performed the PL lifetime measurements of the perovskite films with different contents of IZI in precursor solutions (IZI /PbI₂ molar ratio is x , $x = 1, 2, 3, 4$) (**Supplementary Fig. 1a**). The results show that the PL lifetime increases with the increasing amount of IZI, suggesting that the non-radiative recombination of the γ -CsPbI₃ is suppressed with increasing IZI, which is consistent with the PLQE results (**Supplementary Fig. 1b**). We have added this in the revised manuscript (Line 84 to 88, Page 3, highlighted, **Supplementary Fig. 1**).

Supplementary Fig. 1 Optical properties of CsPbI_3 films on ZnO/PEIE substrate. **(a)** Time-resolved PL for the CsPbI_3 films fabricated from precursor solutions with different mole ratio of IZI and PbI_2 (1:1, 2:1, 3:1, 4:1) under a fluence of 30 nJ cm^{-2} . **(b)** Excitation-intensity-dependent PLQE of CsPbI_3 films fabricated from precursor solutions with different mole ratio of IZI and PbI_2 .

Comment #6. The energy levels such as valence band and conduction band of perovskite samples were not described in this manuscript.

Response: We thank the reviewer for this useful comment. The valence bands of IZI- CsPbI_3 film was obtained via ultraviolet photoelectron spectroscopy (UPS) measurement (**Supplementary Fig. 14**), and the conduction band was estimated using the band gap derived from the absorption band edge. we have added the energy levels of each layer for the device in the revised manuscript (**Fig. 4a**). We have added this in the revised manuscript (Line 206 to 209, Page 8, highlighted, **Supplementary Fig. 14**).

Supplementary Fig. 14 UPS spectra of ZnO/PEIE substrate and IZI- CsPbI_3 film spin-coated on ZnO/PEIE

substrate.

Fig. 4a Flat-band energy level diagram of LED structure. the valence bands of ZnO/PEIE substrate and IZI-CsPbI₃ film was obtained via ultraviolet photoelectron spectroscopy (UPS) measurement, and the conduction band was estimated using the band gap derived from the absorption band edge.

Comment #7: The authors showed a crystal stability test by XRD measurement. The optical stability test of IZI-CsPbI₃ film is also important.

Response: We thank the referee for the helpful suggestion. We have added the evolution of the normalized PL intensity of the CsPbI₃ films with different contents of IZI in precursor solutions (IZI/PbI₂ molar ratio is x, x= 1, 2, 3, 4) on ZnO/PEIE substrate in ambient air. The PL intensity of CsPbI₃ films with x values of 4 dropped to 50% over 8 days, suggesting that excess IZI can effectively enhance their optical stability. (Line 92 to 94, Page 4, highlighted, **Supplementary Fig. 3**);

Supplementary Fig. 3 PL spectrum of CsPbI₃ films fabricated from precursor solutions with different mole ratio of IZI and PbI₂ under 445 nm excitation with light intensity of 12.9 mW cm⁻² in ambient air. (a) 1:1, (b) 2:1, (c) 3:1, (d) 4:1, (e) Evolution of the normalized PL intensity of these films. Note that the PL intensity of CsPbI₃ films with mole ratio of 4 dropped to 50% over 8 day.

Reviewer #2 (Remarks to the Author):

Comment #1: The authors report efficient CsPbI₃ light-emitting diodes by employing an imidazolium additive. The authors have also performed thorough experimental studies to determine the mechanism of low temperature CsPbI₃ perovskite formation through the use of imidazolium. Given that cesium-based perovskites show good promise in long-term device stability, this work is important and could be published in nature communications upon minor revision.

Response: We thank the reviewer for recognising the importance of our work. Guided by these constructive comments, we have made improvements throughout the paper.

Comment #2: Could the authors comment if it is experimentally confirmed that the imidazole remains in the perovskite film/layer? Presumably this could be determined by some spectroscopic techniques.

Response: We appreciate this suggestion. In this work, we detect the chemical structure of the perovskite film by the XPS measurement. two peaks assigned to 401.7 eV (N-3) and 400.2 eV (N-1) of imidazole are observed, confirming the existence of imidazole in the IZI-CsPbI₃ films (**Fig. 2**). ATR-FTIR spectroscopy measurement result can further confirm this result. As shown in

Supplementary Fig. 9a, the signature peak of imidazole (IZ) at about 3130 (CH stretching), 1543 (NH bend), 1328 (CH bend), 1263 (ring breathing), 1055 (CH bend), 841 (ring bend), 757 (CH out-of-plane bend) and 658 (torsion) cm^{-1} are clearly observed.

Fig. 2 XPS spectra of (b) N 1s for IZI, IZI-CsPbI₃ film on ZnO substrate and for IZI, IZ film on ITO substrate, respectively. Confirming the existence of imidazole in the CsPbI₃ films.

Supplementary Fig. 9a ATR-FTIR spectra of IZI, IZ and IZI-CsPbI₃ films on ZnO/PEIE substrate, The signature peak of imidazole (IZ) at about 3345 ($\nu_{\text{N-H}}$), 3130 ($\nu_{\text{C-H}}$), 1543 (δ_{NH}), 1328 (δ_{CH}), 1263 (ring breathing), 1055 (δ_{CH}), 841 (ring bend), 757 (γ_{CH}) and 658 (torsion) cm^{-1} are clearly observed in the CsPbI₃ films.

Comment #2: The authors mentioned that other large ammonium salts worked as well. How crucial is the identity of this ammonium salt for general device performance?

Response: We appreciate the reviewer for raising this interesting point. In this work, we have revealed that the CsPbI₃ black phase could also be fabricated at low temperature through synergistic effect of ZnO and other large organic ammonium salt, such as BAI, HAI, PEAI and NMAI, demonstrating the generality of the approach. The IZI has been successfully used to improve the performance of CsPbI₃ LED by composite engineering and device optimization. We have not made similar effort with other ammonium salts, but we believe it shall be possible to achieve high performance devices with them. The key point we think is to achieve high quality crystals with low defect densities, presumably the ammonium can effectively passivate defects, and to achieve desirable morphology to maximize the light outcoupling efficiency.

Comment #3: The imidazole has helped in the formation of CsPbI₃ at low temperature. Could the authors comment on whether it is helpful for keeping CsPbI₃ in its black phase over the long term? How does it compare to CsPbI₃ formed at high temperature without the help of imidazole?

Response: We have performed the phase stability measurements of the regular γ -CsPbI₃ obtained from the high-temperature annealing process. The γ -CsPbI₃ black phase can only retain for 4 hours in the ambient air (**Supplementary Fig. 2**). In contrast, the IZI-CsPbI₃ film exhibits negligible degradation after exposing for 36 days in the same environment (**Fig. 1a**), suggesting significantly improved phase stability compared to previously reported results. So we believe that the IZI is helpful for keeping CsPbI₃ in black phase for long term. We have added this in the revised manuscript (Line 88 to 91, Page 3, highlighted, **Supplementary Fig. 2**).

Supplementary Fig. 2 XRD pattern of CsPbI₃ film without IZI after exposed in the ambient air for various durations. The film is fabricated on ZnO/PEIE substrate at annealing temperature of 230 °C.

Fig. 1a XRD pattern of IZI-CsPbI₃ film on ZnO/PEIE substrate after exposed in the air with 80 % relative humidity for various durations.

Reviewer #3 (Remarks to the Author):

Comment #1: The authors has developed a low-temperature fabrication method of γ -CsPbI₃, where organic ammonium cations were introduced (through imidazolium iodide (IZI)) to form an intermediate phase to mediate the formation process of the perovskite. Through XPS and ATR-FTIR

characterizations, the authors pointed out that the deprotonation of the IZ^+ cations by ZnO promotes interionic exchange between IZ^+ and Cs^+ , and therefore facilitates the formation of the γ - CsPbI_3 . The work also demonstrated a record EQE of 10.4% in 3D CsPbI_3 -based red PeLEDs. The findings reported in this manuscript are interesting and may have important impact on low-temperature fabrication of thin film all-inorganic perovskites. In general, this is a high-quality work. I would recommend acceptance of this manuscript after the following issues are properly addressed.

Response: We thank the reviewer for well appreciating the importance of our work. We have revised our manuscript carefully according to the reviewer's suggestion.

Comment #2: Figure 2b shows different N 1s peaks of IZI on ZnO and on ITO. It is suggested that ZnO can deprotonate the IZ^+ cation of IZI through formation of Zn-O-H bonds. In principle, ITO should also allow for a similar deprotonation process through formation of Sn-O-H bonding. Why does the latter process not occur? Can the authors please clarify what properties are required for the metal oxide to initiate the deprotonation process?

Response: We think that the chemical acidity of the metal oxide plays a key role in the deprotonation process. The ZnO with a higher isoelectric point (IEP) value of 8.7-10.3 is likely to induce an easier deprotonation of the organic cation. However, the SnO_2 with IEP value of 6.6-9.5 present neutral and difficult react with the organic cation. So we believe that the basic metal oxide substrate could initiate the deprotonation process of organic ammonium cations easily.

Comment #3: Regarding the mechanism illustrated in Figure 2f: if the interionic exchange is enabled by ZnO, which contacts the bottom surface of the perovskite, could the exchange happen efficiently at the top surface of the perovskite? Does the thickness of the perovskite film impose a limit to the effectiveness of this method?

Response: We thank the referee for the insightful comment. Since the ZnO induced deprotonation process likely mainly occurs at the ZnO interface, it is interesting to investigate how thick the perovskite can be formed by this approach. **Supplementary Fig. 11** shows the absorbance of the films with various thickness fabricated from different concentration of precursor solutions. It shows the absorbance at 687 nm from the black phase CsPbI_3 increases linearly with the film thickness less than 200 nm. Above 200 nm, the absorbance saturates and declines. This result suggests that the ZnO

substrate can facilitate the γ -CsPbI₃ formation within the film thickness of 200 nm. We have added this in the revised manuscript (Line 165 to 172, Page 6, highlighted, **Supplementary Fig. 11**).

Supplementary Fig. 11 Effectiveness of ZnO on the thickness of IZI-CsPbI₃ film fabricated with different weight fraction (wt %) of precursor solutions ranging from 10 % to 23 %. (a) thickness of IZI-CsPbI₃ film after annealing, as determined by profilometry. (b) IZI-CsPbI₃ film thickness as a function of the weight fraction (wt %) of precursor solutions. (c) UV-vis spectrum of IZI-CsPbI₃ film. (d) Relationship curve between the absorbance of CsPbI₃ black phase and film thickness.

Comment #4: It might be helpful if the authors could provide some morphological characterizations to assist understanding of the perovskite formation process.

Response: We thank the reviewer for this useful suggestion. The corresponding morphology of IZI-CsPbI₃ films annealed at 100 °C for various time durations is also monitored by SEM measurement (Supplementary Fig. 4). The unannealed film displays a dense, planar morphology (Supplementary Fig. 4a). With a short-time annealing (t=10 s), mounts of small grains of about 40 nm emerge (Supplementary Fig. 4b), corresponding to the intermediate phase. By extending the annealing time duration to 15 s, the small grains grow bigger and the layer becomes discrete

(Supplementary Fig. 4c), corresponding to the mixed phase with 1D and 3D. When annealed over 30 s, the discrete γ -CsPbI₃ grains with an average size of ~ 80 nm form and disperse on the ZnO/PEIE substrate (Supplementary Fig. 4d-f). We have added this in the revised manuscript (Line 109 to 117, Page 4, highlighted, **Supplementary Fig. 4**).

Supplementary Fig. 4 SEM images of IZI-CsPbI₃ film annealed at 100 °C for various time durations on ZnO/PEIE substrate. a) 0 s, b) 10 s, c) 15 s, d) 30 s, e) 60 s, f) 120 s. Scale bar, 400 nm.

Comment #5: For the stage 2 of chemical reaction, Zn²⁺ ions are formed. Will these ions induce trap states?

Response: We are not quite sure about this at the moment. However, previous reports have demonstrated that Zn²⁺ doping in the colloidal CsPbI₃ perovskite can reduce non-radiative charge recombination (*Nano Lett.* 2019, **19**, 1552–1559). So we think the Zn²⁺ ion, which is formed during the process of the IZ⁺ deprotonation by ZnO, might also be useful for decreasing the non-radiative charge recombination to improve the PLQE.

Comment #6: Ammonium salts are reported to passivate defects in perovskite films and at interfaces in PeLEDs, leading to promotion of device performance and lifetime. In this regard, could the addition of IZI also lead to improvement in film quality or interfaces and therefore better LED performances? Also, how about stability?

Response: We thank the reviewer's suggestion. We have performed the excitation-intensity-dependent PLQE and PL lifetime measurements of the perovskite films with

different contents of IZI in precursor solutions (IZI /PbI₂ molar ratio is x , $x= 1, 2, 3, 4$) (**Supplementary Fig. 1**). The results show that the PLQE and the PL lifetime all increases with the x value increasing, suggesting that the non-radiative recombination of the γ -CsPbI₃ is suppressed with increasing IZI. More importantly, the IZI-CsPbI₃ film exhibits negligible degradation after exposing for 36 days in ambient air at room temperature with 80 % relative humidity (**Fig. 1a**). The PL intensity of IZI-CsPbI₃ film dropped to 50% over 8 day in the ambient air (**Supplementary Fig. 3e**), suggesting significantly improved phase and optical stability compared to previously reported results. So the LED based on IZI-CsPbI₃ film (mole ratio of 4: 1.5: 1 for IZI: CsI: PbI₂) exhibits a peak EQE of 10.4 % with suppressed efficiency roll-off under a high current density of 100 mA cm⁻² (**Fig. 4e**). The best device shows a half-lifetime of 20 min at a constant current density of 100 mA cm⁻², and the EL peak position remains constant over time (**Fig. 4f**).

Supplementary Fig. 1 Optical properties of CsPbI₃ films on ZnO/PEIE substrate. **(a)** Time-resolved PL for the CsPbI₃ films fabricated from precursor solutions with different mole ratio of IZI and PbI₂ (1:1, 2:1, 3:1, 4:1) under a fluence of 30 nJ cm⁻². **(b)** Excitation-intensity-dependent PLQE of CsPbI₃ films fabricated from precursor solutions with different mole ratio of IZI and PbI₂.

Fig. 1a XRD pattern of IZI-CsPbI₃ film on ZnO/PEIE substrate after exposed in the air with 80 % relative humidity for various durations.

Supplementary Fig. 3e Evolution of the normalized PL intensity of these films fabricated from precursor solutions with different mole ratio of IZI and PbI₂. Note that the PL intensity of CsPbI₃ films with mole ratio of 4 dropped to 50% over 8 day.

Fig. 4 (e) EQE versus current density (*EQE-J*) curves for CsPbI₃ LEDs. (f) Lifetime and the maximum EL peak position measurement at 100 mA cm⁻². The maximum EQE as the initial value to calculate the T₅₀.

REVIEWERS' COMMENTS:

Reviewer #1 (Remarks to the Author):

The authors revised the manuscript carefully the reviewer suggestions. I believe this work would draw interest in the perovskite LEDs a publication in Nature Communications.

Reviewer #2 (Remarks to the Author):

The authors have addressed the reviewers' comments and I am happy to recommend acceptance for publication.

Reviewer #3 (Remarks to the Author):

The authors have properly addressed all my comments. I would like to recommend acceptance of this manuscript.